# A Real-World Cost-Effectiveness Study Evaluating Imaging Strategies for the Diagnostic Workup of Renal Colic in the Emergency Department

**DOI:** 10.3390/medicina59030475

**Published:** 2023-02-28

**Authors:** Sabrina Kepka, Kevin Zarca, Mickaël Ohana, Anne Hoffmann, Joris Muller, Pierrick Le Borgne, Emmanuel Andrès, Pascal Bilbault, Isabelle Durand Zaleski

**Affiliations:** 1Emergency Department, Hôpitaux Universitaires de Strasbourg, 1 Place de L’hôpital, CHRU of Strasbourg, 67091 Strasbourg, France; 2ICUBE UMR 7357 CNRS, Équipe IMAGeS, 300 Bd Sébastien Brant, 67400 Illkirch-Graffenstaden, France; 3Assistance Publique-Hôpitaux de Paris, DRCI-URC Eco Ile-de-France, 1 Place du Parvis Notre Dame, 75004 Paris, France; 4Assistance Publique-Hôpitaux de Paris, Service de Santé Publique, Henri Mondor-Albert-Chenevier, 94000 Créteil, France; 5Radiology Department, Nouvel Hôpital Civil, Hôpitaux Universitaires de Strasbourg, 1 Place de L’hôpital, 67091 Strasbourg, France; 6Public Health Unit, Hôpitaux Universitaires de Strasbourg, 1 Place de L’hôpital, CHRU of Strasbourg, 67091 Strasbourg, France; 7UMR 1260, INSERM/Université de Strasbourg CRBS, 1 rue Eugene Boeckel, 67000 Strasbourg, France; 8Department of Internal Medicine, Hôpitaux Universitaires de Strasbourg, 1 Place de L’hôpital, 67091 Strasbourg, France; 9CRESS, INSERM, INRA, Université Paris-Est Créteil (UPEC), 94000 Créteil, France

**Keywords:** tomography, ultrasound, emergency department length of stay, cost-effectiveness analysis, renal colic, urolithiasis

## Abstract

***Introduction*** Both non-contrast Computed Tomography (CT) and ultrasound (US) are used for the diagnosis of renal colic in the emergency department (ED). Although US reduces radiation exposure, its diagnostic accuracy is inferior to that of CT. In this context, data regarding the cost and organizational impact of these strategies represent essential elements in the choice of imaging; however, they remain poorly documented. ***Aim of the study*** The aim of this study was to compare the costs and effectiveness of diagnostic workup by US and CT for patients consulting with renal colic in the ED. ***Methods*** We conducted a monocentric real-life retrospective study of patients consulting for a renal colic in an ED between 1 July 2018 and 31 December 2018. We estimated length of stay (LOS), total hospital costs at 60 days including ED, and initial and repeat admissions. Patients with initial US in the ED were compared to patients with initial CT using inverse probability weighting of the propensity score calculated from demographic variables, vital parameters, and clinical presentation. We calculated the incremental cost effectiveness ratio as the difference in costs by the difference in LOS. The variability of the results was assessed using non-parametric bootstrapping. ***Results*** In this study, of the 273 patients included, 67 were patients assessed with US and 206 with CT. The average costs were €1159 (SD 1987) and €956 (SD 1462) for US and CT, respectively, and the ED LOS was 8.9 [CI 95% 8.1; 9.4] and 8.7 [CI 95% 7.9; 9.9] hours for US and CT, respectively. CT was associated with a decreased LOS by 0.139 [CI 95% −1.1; 1.5] hours and was cost-saving, with a €199 [CI 95% −745; 285] reduction per patient. ***Conclusion*** When imaging is required in the ED for suspected renal colic as recommended, there is real-life evidence that CT is a cost-effective strategy compared to US, reducing costs and LOS in the ED.

## 1. Introduction

Renal colic is a common disease. Due to many factors such as change in diet or climate, the prevalence of urolithiasis has increased since 1980 in the United States as well as in other countries, from 3% to 10% in 2016 with stable hospitalization rates [1,2]. Therefore, the diagnostic workup and management of renal colic primarily involves the emergency departments (ED). An estimated two million ED visits occur annually in the United States, with costs reaching $5 billion [3,4,5].

Both non-contrast Computed Tomography (CT) and ultrasound (US) are used for the diagnosis of renal colic. Non-contrast CT of abdomen and pelvis has high sensibility and specificity for the detection of urinary stones and is considered the gold standard [6,7]. Low-dose CT is a relevant alternative with a pooled sensitivity between 93% and 94%, and a specificity between 88 and 100% in two meta-analyses using standard dose CT as a reference [8,9]. US reduces radiation exposure, which may be important because of the young age of patients with high recurrence rates; however, its accuracy is lower than that of CT and low-dose CT [10]. A review confirmed the moderate diagnosis accuracy, detecting hydronephrosis with a sensibility from 72% to 87% and a specificity from 73% to 83%, given that the sensitivity and specificity of hydronephrosis to detect a stone are 80% and from 37% to 78%, respectively [11]. The pooled results from point-of-care ultrasound yielded a sensitivity of 70.2% and specificity of 75.4%, although the specificity increases to 94.4% when moderate or greater hydronephrosis is used as a criterion [12].

In addition to diagnostic accuracy and organizational benefits, health care consumption and costs should be considered as relevant consequences of imaging performed in the ED. Yet, there are few data on costs and the organizational impact of CT vs. US in the evaluation of suspected renal colic [10].

Therefore, our objective was to assess the cost-effectiveness of CT compared to US in the ED for the diagnostic workup of patients with renal colic.

## 2. Materials and Methods

### 2.1. Study Population, Setting and Location

We conducted a retrospective monocentric study in the ED of a university hospital in France.

### 2.2. Comparators

Patients were divided into two groups according to the initial abdominal imaging requested in the ED, “US” or “CT,” under real-life conditions according to the judgment of the emergency physicians and radiologists in charge of the patient. Subsequent patient management and referral of the patient was performed by the emergency physicians according to the clinical presentation and the results of imaging and laboratory tests.

### 2.3. Perspective

The analysis was conducted from the perspective of the French healthcare provider using hospital production costs or proxies (tariffs) when costs were not available.

### 2.4. Time Horizon

The time horizon was 60 days after the discharge of the hospital or ED.

### 2.5. Discount Rate

No discount rate was applied.

### 2.6. Measure of Effectiveness

We evaluated the impact of initial imaging choice on ED length of stay (LOS). LOS was measured as the time from ED admission to ED discharge (discharge or transfer to a medical service), as recorded in the ED electronic medical record.

### 2.7. Measurement and Valuation of Resources and Costs

Data for the economic evaluation were collected retrospectively.


Costs items:
-Number and unit costs of US and CT performed in the ED;-Costs of initial hospitalization and readmissions related to initial renal colic during the 60-day follow up period.


### 2.8. Currency, Price Date, and Conversion

Hospitalization and monitoring data were obtained from local hospitals claims databases. Hospitalizations were valued using the corresponding French disease-related group (DRG) cost, adjusted by LOS and number of days in intensive care [13,14,15]. For patients who were not hospitalized, we imputed the average cost of outpatient ED visits plus the cost of ED imaging, assessed using the statutory health insurance tariffs (Appendix A).

### 2.9. Selection, Measurement and Valuation of Effectiveness

#### Primary Endpoint

The incremental cost-effectiveness ratio (ICER) was calculated as the difference in costs divided by the difference in ED LOS.

As the secondary endpoints, we measured:Time to imaging, measured as the time from admission to the time of first imaging, as recorded in the ED electronic health record;Proportion of patients who underwent a second imaging in the ED;Care pathway after ED workup as a proportion of admissions (by medical service type, including intensive care unit [ICU] and short-stay unit);Proportion of rehospitalizations during follow-up from local hospital claims databases;Proportion of patients with imaging during 60-day follow-up recorded in medical file.

### 2.10. Rationale and Description of Model

In order to compare the outcomes between US and CT in the EDs in this non-randomized study, we used a regression model weighted by inverse probability of the propensity score using demographic variables (age and sex), laboratory tests (C Reactive Protein-CRP-, leucocyte numbers, and Cockcroft estimated glomerular filtration rate-eGFR-), and presence of hydronephrosis.

For missing data, we used multiple imputation with chained equations to create and analyze 10 multiply imputed datasets. Incomplete variables were imputed according to a fully conditional specification, using the default settings of the mice 3.0 package [16]. For each data set, we used a generalized linear model to model ED LOS and employed a Gaussian family with a log link because the distribution of ED LOS was skewed and lower bounded to 0. For the costs, we used a generalized linear model with a gamma family with a log link. We combined estimates on each imputed data set using Rubin’s rules.

### 2.11. Analytics and Assumptions

We computed the groups’ marginal effectiveness and costs with US as a reference. For the main effectiveness measure, we used the negative value of LOS in the ED (2 h became −2 h) in order to reflect the incremental cost saved for 1 h saved in the ED.

Quantitative variables were described as mean ± standard deviation, while categorical variables were described as numbers and percentages. Chi-square or Fisher’s exact tests, analysis of variance (ANOVA), and the Kruskal–Wallis test were used as appropriate. There were no missing data concerning cost and effectiveness since we extracted the data from the electronic health records and the health insurance national database. A *p*-value < 0.05 was considered significant.

### 2.12. Characterizing Uncertainty

The variability of the results was assessed using non-parametric bootstrapping, which provided multiple estimates of the ICER by randomly re-sampling the patient population 1000 times. For each iteration, we computed propensity scores and new regression models. The results are presented as a scatter plot of 1 000 ICERs on the cost-effectiveness plane. 

### 2.13. Characterizing Heterogeneity

As health care costs in France are generally lower than in other countries, especially the United States, we performed a deterministic sensitivity analysis to investigate the independent effect of the following variables on the ICER by substituting items of cost with five times the base case for each parameter.

All analyses were performed using R software version 4.0.3. (R Development Core Team 2020).

### 2.14. Ethics Approval, Data and Safety Monitoring

This study was conducted in accordance with the principles set forth by Good Clinical Pratice guidelines and the declaration of Helsinski. The study was approved by the Ethics Committee (CE 2019-94). A declaration of conformity was obtained from the *Commission nationale de l’informatique et des libertés* (CNIL) (agreement number 2208067v0). In accordance with French legislation, formal written informed consent was not required for this type of study because the data were entirely retrospectively studied [17].

## 3. Results

A total of 273 consecutive patients aged over 18 years consulting for renal colic with US or CT performed in the ED for the diagnostic workup were included from 1 July 2018 to 31 December 2018 (Figure 1).

The majority (N = 206) of the 273 patients underwent a CT scan, and an US was performed for 67 patients. The mean age was 46.5 (±15.4) in the “CT group” and 47.4 (±17.1) in the “US group” (*p* = 0.69) (Table 1). No differences were found between the groups with respect to medical history and clinical presentation. In this cohort, eGFR was significantly higher in the “US group,” as well as CRP, than in the “CT group”: respectively, 94.5 versus 88.4 and 8.1 versus 13.9 (*p* = 0.02 and 0.04).

### 3.1. Study Parameters

#### 3.1.1. Effectiveness

The ED LOS was non-significantly shorter when a CT was initially performed in the ED, with a mean LOS of 8.6 (±3.7) hours versus 8.8 (±4.5) hours for US (*p* = 0.32) in the unadjusted population.

#### 3.1.2. Secondary Endpoints

The time to imaging was not significantly shorter in the “CT group” than in the “US group” (*p* = 0.23).

A second imaging was performed for 18% of patients after initial US (*p* < 0.01).

More patients were hospitalized in the “CT group” than in the “US group” (*p* < 0.01), and they were mainly in the short time unit (*p* = 0.02).

During the 60-day follow up period, rehospitalization occurred for 8.7% patients in the “CT group” versus 8.8% in the “US group”.

Finally, more imaging was performed during the follow-up after hospital discharge in the US group (*p* < 0.01).

### 3.2. Incremental Costs and Outcomes

#### 3.2.1. Costs

The average 60-day unadjusted costs in euros were estimated to be €1159 (±1987) and €956 (±1462) for the “US group” and “CT group,” respectively (Table 2). The most important difference in costs between the strategies concerns rehospitalizations related to initial management: €428 (±1809) and €235 (±1153), respectively, for US and CT.

#### 3.2.2. Group Comparisons

A graph (Figure 2) showing the balance of covariates was constructed to demonstrate the balancing effect of weighting. An SMD (standardized mean differences) value of less than 0.1 is often considered a good signal of the absence of imbalances between groups. We see that the inverse probability weighting effectively corrects apparent imbalances in these variables, leading to a better standardized difference in means for the variables included in the logistic model. Indeed, all of them were under 0.1 after weighting, shown in blue on the graph (Figure 2).

The average LOS in the ED and the average cost in euros with the inverse probability weighting are presented in Table 3. The difference in cost was € − 199 and the difference in efficacy was 8.4 min. The point estimate of the ICER (the difference in costs divided by the difference in LOS in the ED) was € − 1431 (saved)/for a one-hour gain in efficacy (LOS in the ED) by the “CT group” compared with the “US group” (Table 3).

### 3.3. Characterizing Uncertainty

The set of ICERs estimated by the non-parametric bootstrap are presented on the cost-effectiveness plane, with “US” as the reference (Figure 3). About 50% of these replications were located in the bottom right-hand quadrant, indicating a 50% probability of dominance in favor of CT over US, corresponding to a lower cost for greater effectiveness for CT (Figure 3).

### 3.4. Characterizing Heterogeneity

In the deterministic sensitivity analysis (Table 4) which was conducted to investigate the independent effect of the following variables on the ICER by substituting items of cost with five times the base case for each parameter, CT remained dominant in all of the analyses, even with a unit cost of imaging that was five times the cost in France and substituting hospitalizations costs.

## 4. Discussion

In this retrospective study of 273 patients diagnosed with renal colic, the performance of CT as initial imaging for patients presenting in the ED with suspected ureteral stone reduced ED LOS by a mean of 0.139 [CI 95% −1.1; 1.5] hours and was cost-saving, with a reduction of € − 199 [CI 95% −745; 285] per patient and a 50% probability of dominance in favor of CT over US.

Although there is controversy regarding the radiation risk of CT, most practitioners and organizations continue to adhere to the “as low as reasonably achievable” principle for radiation levels in diagnostic imaging. The American College of Emergency Physicians, the American College of Radiology, and the American Urological Association recommend the use of reduced-dose CT [18,19,20]. As well, based on a systematic review of the literature, a Delphi process yielded expert consensus on optimal imaging in 29 specific clinical scenarios, supporting ultrasound or no imaging in specific clinical scenarios, with reduced-dose radiation CT to be used when CT is needed for patients with suspected renal colic [12]. One study evaluated the implementation of an audit assessing the degree of CT imaging use and then implementing measures to reduce unnecessary radiation exposure [21]. In this study, CT was often used as the initial diagnostic modality for suspected recurrent renal colic. The study commissioned a new low-dose CT in this ED, which is specifically aimed at reducing radiation exposure in patients that are likely to undergo repeated examinations.

Thus, when imaging is required, the choice of imaging depends on a number of factors, including patient characteristics and clinical status. This choice also has an impact on efficient triage in the ED, since there is a controlled LOS, which is crucial in the context of ED overcrowding involving adverse events [22,23,24,25,26,27]. However, the impact of imaging workup for renal colic on ED LOS remains a topic of debate. Indeed, point-of-care US could reduce ED LOS compared to CT; however, it could increase LOS when a second imaging must be performed, as in the case of US performed by a radiologist [28,29,30].

In addition, the management of ureterolithiasis is the most expensive of the most common outpatient visits after ED, primarily due to imaging costs [31]. Only five studies have evaluated the costs of an ED visit for urolithiasis in the United States, and they suggested that patients should pay between $2300 and $6000 [30,31,32,33,34]. When considering only the cost in the ED, US was less costly than CT, with a very small difference from $319 to $423 versus from $259 to $449, respectively, likely due to the cost difference regarding imaging [30,34]. However, despite the low hospitalization rate of only 11% of patients, hospital admissions contributed to more than 50% of the total costs, estimated at $970 for radiologist US and at $959 for CT [34].

After ED management, the consumption of care is another criterion to be considered for the effectiveness of the imaging strategy. However, there are few data on ED revisits as a function of imaging strategy and results concerning ED revisits differ according to the studies. In the study of Blecher et al., revisits to the ED were less frequent for US than CT, with rates of 12% and 19%, respectively; however, a second CT was performed in the ED for more than 50% of patients with initial US [29]. Although other studies have not found a difference in revisits according to initial imaging [28,35], our study revealed that the strategy involving US was more costly than CT, particularly because of the cost of rehospitalization and the need to perform a second imaging in the ED for some patients. In addition, CT can be used to make diagnoses other than ureteral stone, especially with low-dose CT, which reduces radiation while maintaining a better diagnosis accuracy than US [36].

### Limitations and Strengths

In this retrospective study, other factors than the strategy applied for the diagnostic workup could explain the differences between the groups. However, we performed an inverse probability weighting from propensity scores to reduce heterogeneity between the groups.

Then, our cost calculations considered ED costs and subsequent hospital costs during a 60-day follow-up period without outpatient costs, because hospitalization accounted for the majority of strategy-related costs.

Another limitation concerns the generalization of the results, given that costs differ between countries. The average costs in the Melkinov study were $141 for US performed by a radiologist and $248 for CT, respectively [34]. To account for these differences between health systems, we performed a deterministic sensitivity analysis and CT remained cost-effective in all settings.

Finally, our measure of effectiveness did not take into account diagnostic accuracy and patients’ health states, which could be debated. However, the superiority of CT diagnostic accuracy over US was already known, and the goal of changing the imaging strategy in the ED was correlated with the need for efficient triage to optimize patient flow. Indeed, the LOS in the ED is notably influenced by the imaging strategy and represents a major outcome in evaluating the effectiveness of imaging on the management of patients in these units, with the need for quick and accurate decisions to initiate treatment if necessary and control overcrowding. Thus, the time spent in the ED seemed to be the better performance outcome.

To our best knowledge, this is the first study to provide real-world data on the cost-effectiveness of imaging strategies performed in the ED for the diagnostic workup of patients with renal colic. These results provide important data for health care payers, as CT was the more cost-effective strategy compared with US. Thus, there is real-world evidence that CT is less expensive than US with a high probability. Although there is an advantage to US, particularly for pregnant patients, CT appears to be an efficient strategy compared with US, with organizational benefits for EDs by reducing LOS, the number of imaging tests to be performed in these overcrowded units, and overall costs. However, the challenge for clinicians remains to identify patients for whom imaging is strictly necessary in order to reduce radiation exposure of patients who are often young and have a high recurrence rate. In this context of the need for a reasoned use of CT, decision support tools have been proposed to facilitate shared decision making, the effectiveness of which should be evaluated in clinical practice in the ED [37]. The availability of CT, especially low-dose CT, is a relevant method to improve patient flow in the ED with a rapid care pathway dedicated to suspected ureteral calculi. In our center, for example, multidisciplinary collaboration between radiologists and emergency physicians has made it possible to set up a care network with rapid access to the ulta-low-dose CT for patients consulting with a strong clinical suspicion of renal colic, as soon as they arrive in the ED (excluding women of childbearing age). This may be a solution to allow for more efficient management of these patients in the ED. Finally, it would be very informative to compare these data with a more recent dataset, as the trend and difference between the two modalities (US and CT) would be even more significant with the increasing availability of low-dose CT scanners allowing reduced radiation with better diagnostic relevance compared to US.

## 5. Conclusions

Our real-life study showed that a CT diagnostic workup for renal colic reduced the length of stay in the ED by 8 min per patient, and most importantly, reduced the average cost by 199 € per patient compared to ultrasound. However, the main limitation of CT remains radiation exposure, especially in young patients or pregnant women. Thus, it is essential to limit the use of imaging to patients for whom it is strictly necessary and to encourage the use of LD CT in accordance with recommendations. Strategies to increase access to ultra-low dose CT, outside of contraindications, would improve patient management, reduce costs, and have organizational benefits in reducing emergency room overcrowding.

## Figures and Tables

**Figure 1 medicina-59-00475-f001:**
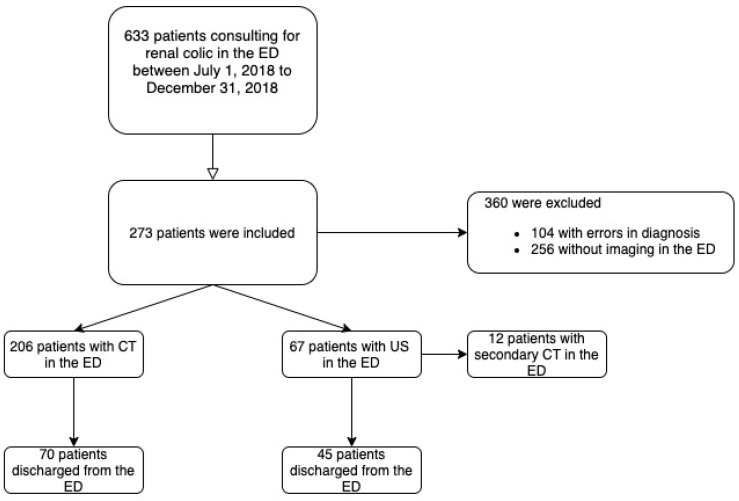
Flowchart of the study conducted in the ED of a University Hospital in France. ED: emergency department. CT: computed tomography.

**Figure 2 medicina-59-00475-f002:**
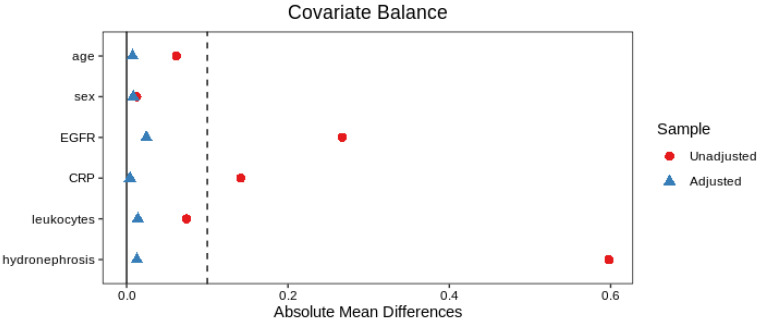
Standardized mean difference (SMD) in the unadjusted and adjusted with propensity score populations. CRP: C Reactive Protein. eGFR: Cockcroft estimated glomerular filtration rate.

**Figure 3 medicina-59-00475-f003:**
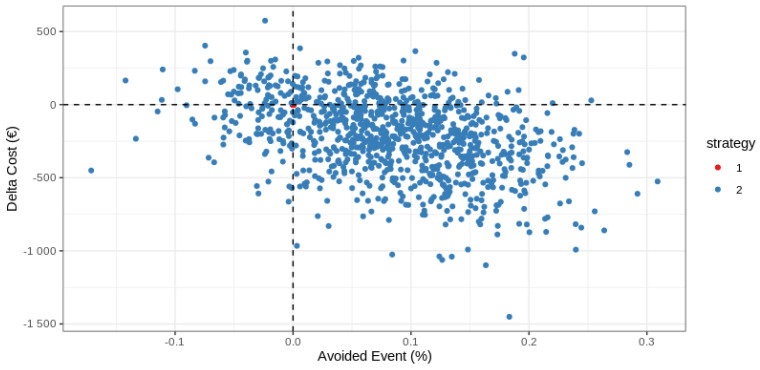
Scatter plot of incremental cost and effectiveness of CT (strategy 2) compared to US (strategy 1) as the reference.

**Table 1 medicina-59-00475-t001:** Cohort characteristics, length of stay in the ED, and outcome.

Characteristics		US N = 67	CT N = 206	*p*
Age (M, SD)		47.4 (±17.1)	46.5 (±15.4)	0.69
Gender				
	Men (N, %)	48 (72)	145 (70)	0.84
	Women (N, %)	19 (28)	145 (70)	
Medical history				
	Nephrolithiasis (N, %)	28 (42)	66 (32)	0.14
	Renal malformation (N, %)	2 (3)	7 (3.4)	0.25
Clinical presentation				
	Heart rate (M, SD)	78.6 (±13.5)	77 (±15.1)	0.42
	Systolic blood pressure (mmHg) (M, SD)	140 (±19)	140 (±19.8)	1
	Pain numerical scale * (Median, Q25-75)	8 [6; 10]	8 [6.7; 10]	0.76
	Body temperature (M, SD)	36.9 (±0.7)	36.7 (±0.6)	0.06
Laboratory tests				
	Leukocytes (M, SD)	10.4 (±4)	10.6 (±3.4)	0.72
	Cockcroft estimated glomerular filtration rate (µmol/L) (M, SD)	94.5 (±23.8)	88.4 (±23.6)	**0.02**
	CRP (M, SD)	8.1 (±16.2)	13.9 (±39.9)	**0.04**
Hydronephrosis ** (M,SD)		11.7 (±9.1)	16.2 (±6.7)	**<0.01**
Time to imaging in the ED * (Median, Q25-75)		4.5 [3.4; 5.9]	4.8 [3.7; 6.8]	0.23
Another imaging in the ED		12 (18)	0 (0)	**<0.01**
Length of stay in the ED * (M, SD)		8.8 (±4.5)	8.6 (±3.7)	0.32
Care pathway after ED				
	Discharge from the ED (N, %)	45 (67)	70 (34)	**<0.01**
	Hospitalization in the short time stay unit (N, %)	19 (28)	92 (45)	**0.02**
	Hospitalization in urology unit (N, %)	3 (5)	44 (21)	**<0.01**
Length of hospitalization ‡ (Median, Q25-75)		1 [1;1]	1 [1;1]	1
Another unplanned imaging during follow-up at 60 days		14 (20.8)	14 (6.8)	**<0.01**
Death at 60 days (N, %)		0 (0)	0 (0)	1

*: hours. ‡: days. *: numerical pain from 0 to 10. **: mm. ED: emergency department. Q: quartiles. CT: computed tomography. US: ultrasound.

**Table 2 medicina-59-00475-t002:** Average costs per patient in € in the group ultrasound and CT in the ED for the unadjusted population.

Costs Expressed as Mean (SD)	US N = 67	CT N = 206
Imaging	€43.7 (±49.6)	€29.6 (±38.2)
Consultation in the ED	€33.4 (±39.8)	€30.3 (±38.9)
Hospitalization	€649 (±781)	€660 (±826)
Reconsultation in the ED	€4.5 (±25.7)	€1.5 (±14.7)
Rehospitalization	€428 (±1809)	€235 (±1153)
Total cost	€1159 (±1987)	€956 (±1462)

ED: emergency department. CT: computed tomography. US: ultrasound.

**Table 3 medicina-59-00475-t003:** Costs and effectiveness for each strategy of imaging (weighted endpoints).

Strategy [95% CI]	LOS in the ED	Cost Per Patient	Difference of LOS in the ED	Difference of Cost
US	8.9 [8.1; 9.4]	€1159 [745; 1247]	NA	NA
CT	8.7 [7.9; 9.9]	€961 [779; 1680]	0.139 [−1.1; 1.5]	€ − 199 [−745; 285]

LOS: Length of stay presented in hours. ED: emergency department. LOS is presented in hours. CT computed tomography. US ultrasound.

**Table 4 medicina-59-00475-t004:** Deterministic sensitivity analysis.

Determinant of Analysis	Difference of Cost	Difference of_Efficacy
Base case	€ − 199	0.2
Imaging	€ − 253	0.2
Outpatient visit	€ − 219	0.2
Hospitalization	€ − 918	0.2

LOS: length of stay. ED: emergency department.

## Data Availability

Not available.

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
