# Peer review of "A Real-World Cost-Effectiveness Study Evaluating Imaging Strategies for the Diagnostic Workup of Renal Colic in the Emergency Department"

_medicina, 2023, doi:10.3390/medicina59030475_

Round 1

Reviewer 1 Report

       the manuscript is certainly interesting, and the topic is very relevant. There are potentially very valuable lessons to learn from this manuscript. However, many issues need to address before the manuscript is suitable for publication.

Overall:
1. There are many spelling errors in this manuscript. See guide for authors for a free grammar checker. E.g.……..etc. 

Title: - "Cost effectiveness of imaging strategies for the diagnostic workup of renal colic in the emergency department: a real-life retrospective study."

1.Title should be a brief phrase describing the contents of the paper.

Abstract often composed with five parts including Introduction, aim, methods, results, and conclusion. The background should be integrated into aim.: -
1. Introduction: Please describe the problem or lack of knowledge that address by this study.

2. Aim of the study: relevant.
3. Methods: relevant.

4. The result; - should be presented with clarity and precision.
5. The conclusion good
6. The keywords are following the abstract, use about five to 10 key words do not mention in the title so please changes these words without any abbreviations.

Introduction
1. The introduction is ideally in structures.

Methods

1.The methods section written in a way that everyone could repeat this study in the same manner.

Results
This is very important as well. You may collect enough data but be concise and only report essential RESULTS.
2. Results are meaningless without a better description in figures and tables. So please add this and provide a thorough analysis of these results.

Discussion

1.A discussion should offer a short overview of the results, and an in-depth discussion of the interpretation of them.
2. Do the authors have an explanation on why the results are different compared to other studies?
3. The last paragraphs of the discussion are very relevant. Please add some suggestions on how such problems can improve in your own hospital. How did you manage other problems? Share some solutions that applied after this research, so that other hospitals with similar problems can benefit from this research.

Conclusion:
is logic in manner and relevant.
References:
1. Please use uniform references, when available with DOI.

2. Make sure update the old references

Author Response

Revisions manuscript medicina-2190111

Cost effectiveness of imaging strategies for the diagnostic workup of renal colic in the emergency department: a real-life retrospective study

Responses to Reviewers' Comments

Reviewer 1

The manuscript is certainly interesting, and the topic is very relevant. There are potentially very valuable lessons to learn from this manuscript. However, many issues need to address before the manuscript is suitable for publication.Overall:
1. There are many spelling errors in this manuscript. See guide for authors for a free grammar checker. E.g.……..etc.  

Response: We made the changes in the text.

Title: - "Cost effectiveness of imaging strategies for the diagnostic workup of renal colic in the emergency department: a real-life retrospective study."

1.Title should be a brief phrase describing the contents of the paper.

Response: we thank the reviewer and have changed the title to “ A real-world cost-effectiveness study evaluating imaging strategies for the diagnostic workup of renal colic in the emergency department”

Abstract often composed with five parts including Introduction, aim, methods, results, and conclusion. The background should be integrated into aim
1. Introduction: Please describe the problem or lack of knowledge that address by this study. 

  1. Aim of the study: relevant.
    3. Methods:relevant.
  2. The result; -should be presented with clarity and precision.
    5. The conclusiongood
    6. The keywords are following the abstract, use about five to 10 key words do not mention in the title so please changes these words without any abbreviations.

Response: We thank the reviewer for his comments and we modified the abstract as follows:

“Introduction

Both non contrast Computed Tomography (CT) and ultrasound (US) are used for the diagnosis of renal colic in the emergency department (ED). Although US reduces radiation exposure, the diagnostic accuracy is inferior to that of CT. In this context, data regarding the cost and organizational impact of these strategies represent essential elements in the choice of imaging but remain poorly documented.

Aim of the study

The aim of this study was to compare the costs and effectiveness of diagnostic workup by US and CT for patients consulting with renal colic in the ED.”

Results

In this study, of the 273 patients included, 67 patients assessed with US and 206 with CT. The average costs were €1159 (SD 1987) and €956 (SD 1462) for US and CT respectively and the ED LOS was 8.9 [CI 95% 8.1; 9.4] and 8.7 [CI 95% 7.9; 9.9] hours for US and CT respectively. CT was associated with a decrease LOS of 0.139 [CI 95% -1.1; 1.5] hours and was cost-saving with a -199 [CI 95% -745; 285] reduction per patient.”

We modified the key words as follows:

Key words

Tomography

Ultrasound

Emergency department length of stay

Cost-effectiveness analysis

Renal colic

Urolithiasis

Introduction
1. The introduction is ideally in structures.

Methods

1.The methods section written in a way that everyone could repeat this study in the same manner. 
Response: We thank the reviewer for appreciating our work

Results
This is very important as well. You may collect enough data but be concise and only report essential RESULTS. 
2. Results are meaningless without a better description in figures and tables. So please add this and provide a thorough analysis of these results.

Response: We thank the reviewer for this remark and we have better detailed the description of the figures and tables

Discussion

1.A discussion should offer a short overview of the results, and an in-depth discussion of the interpretation of them. 

  1. Do the authors have an explanation on why the results are different compared to other studies?

Response: Few studies have evaluated the rehospitalization, with results concerning emergency department revisits differing according to the studies. But we found that in a number of cases, a second imaging was performed in the emergency department during initial management, which is consistent with other studies. We added a sentence in the discussion as follows:

Although other studies have not found a difference in revisits based on initial imaging [28] [35], our study revealed that the strategy involving US was more costly than CT, particularly because of the cost of rehospitalization and the need to perform a second imaging in the ED for some patients.”

  1. The last paragraphs of the discussion are very relevant. Please add some suggestions on how such problems can improve in your own hospital. How did you manage other problems? Share some solutions that applied after this research, so that other hospitals with similar problems can benefit from this research.

Response: We thank the reviewer for this very pertinent remark. Indeed, we have added a proposal implemented in our hospital to facilitate access to the CT and improve the management of these patients that could be useful to implement in other centers. We added in the discussion:

In our center, for example, a multidisciplinary work between radiologists and emergency physicians has made it possible to set up a care network with rapid access to the ultra-low-dose CT for patients consulting with a strong clinical suspicion of renal colic, as soon as they arrive in the ED (excluding women of childbearing age). This may be a solution to allow for more efficient management of these patients in the ED.”

Conclusion:
is logic in manner and relevant.

Response: Again, we thank the reviewer for appreciating our work
References: 
1. Please use uniform references, when available with DOI.

  1. Make sure update the old references

Response: We have updated the references and added DOI.

Reviewer 2 Report

The authors conducted a single-centre retrospective study between the 1st of July 2018 and 31st of December 2018 to compare the costs and effectiveness of using ultrasound or computer tomography in the diagnosis of patients presenting with renal colic to the emergency department. During the 6-month period the authors included 273 consecutive patients.

The manuscript adequately describes the recruitment process and gives detailed information regarding the 2 cohorts of patients with relevant statistical parameters. The mathematical and statistical formulae are well-explained and adequate for the relevant type of datasets.

In the limitations section, the paper describes several relevant limitations; however, the authors fail to elaborate on why they were using data from 2018 instead of a more recent dataset. In Table 1, "Another imaging during follow-up at 60 days" could be further divided into planned and unplanned sections.

This is a very informative manuscript based on real-life data. The authors appropriately attempted to compensate for any confounding factors emerging from the uncertainties of unscheduled care. It would be very informative to compare the data with a more up-to-date dataset as I suspect the trend and difference between the two modalities (US and CT) would be even more significant. 

Please run a detailed spelling/grammar check on the manuscript  ("input" spelt as "imput" multiple times in the "Rationale and description of model" section)

Author Response

Revisions manuscript medicina-2190111

Cost effectiveness of imaging strategies for the diagnostic workup of renal colic in the emergency department: a real-life retrospective study

Responses to Reviewers' Comments

Reviewer 2

The authors conducted a single-centre retrospective study between the 1st of July 2018 and 31st of December 2018 to compare the costs and effectiveness of using ultrasound or computer tomography in the diagnosis of patients presenting with renal colic to the emergency department. During the 6-month period the authors included 273 consecutive patients.

The manuscript adequately describes the recruitment process and gives detailed information regarding the 2 cohorts of patients with relevant statistical parameters. The mathematical and statistical formulae are well-explained and adequate for the relevant type of datasets.

In the limitations section, the paper describes several relevant limitations; however, the authors fail to elaborate on why they were using data from 2018 instead of a more recent dataset.

Response: We evaluated 2018 data retrospectively before the implementation of a rapid access to CT for renal colic management to assess the benefits of CT and ultrasound. With the covid outbreak, we took time to evaluate the results. It will be interesting to evaluate the results of this fast access to CT following the multidisciplinary work carried out between radiologists and emergency physicians in our center. We added in the discussion section as follows:

In our center, for example, a multidisciplinary work between radiologists and emergency physicians has made it possible to set up a care network with rapid access to the ulta-low-dose CT for patients consulting with a strong clinical suspicion of renal colic, as soon as they arrive in the ED (excluding women of childbearing age). This may be a solution to allow for more efficient management of these patients in the ED.”

In Table 1, "Another imaging during follow-up at 60 days" could be further divided into planned and unplanned sections.

Response: These imagings were just unplanned imaging. We have added it in the table.

This is a very informative manuscript based on real-life data. The authors appropriately attempted to compensate for any confounding factors emerging from the uncertainties of unscheduled care. It would be very informative to compare the data with a more up-to-date dataset as I suspect the trend and difference between the two modalities (US and CT) would be even more significant. 

Response: Indeed, the reviewer is right to point out that it would be very informative to compare these data with a more recent data set, as the trend and difference between the two modalities (US and CT) would be even more significant with the increasing availability of low-dose CT scanners allowing for reduced radiation with better diagnostic relevance compared to ultrasound. We added in the discussion as follows:

Finally, it would be very informative to compare these data with a more recent dataset, as the trend and difference between the two modalities (US and CT) would be even more significant with the increasing availability of low-dose CT in ED allowing reduced radiation with better diagnostic relevance compared to US.”

Please run a detailed spelling/grammar check on the manuscript ("input" spelt as "imput" multiple times in the "Rationale and description of model" section)

Response: We have corrected the grammatical errors. However, imputation should be considered in the sense of allocation and it seems to me that in this sense it is written "imput".